# Predicting Active Sites in Photocatalytic Degradation Process Using an Interpretable Molecular-Image Combined Convolutional Neural Network

Zhuoying Jiang [1,†], Jiajie Hu [2,†], Anna Samia [3] and Xiong (Bill) Yu [1,2,4,*]

[1] Department of Civil and Environmental Engineering, Case Western Reserve University, 2104 Adelbert Road, Cleveland, OH 44106, USA; zxj45@case.edu
[2] Department of Electrical, Computer and Systems Engineering, Case Western Reserve University, 2104 Adelbert Road, Cleveland, OH 44106, USA; jxh919@case.edu
[3] Department of Chemistry, Case Western Reserve University, 2104 Adelbert Road, Cleveland, OH 44106, USA; axs232@case.edu
[4] Department of Computer and Data Science, Case Western Reserve University, 2104 Adelbert Road, Cleveland, OH 44106, USA
[*] Correspondence: xxy21@case.edu
[†] These authors contributed equally to this work.

**Abstract:** Machine-learning models have great potential to accelerate the design and performance assessment of photocatalysts, leveraging their unique advantages in detecting patterns and making predictions based on data. However, most machine-learning models are "black-box" models due to lack of interpretability. This paper describes the development of an interpretable neural-network model on the performance of photocatalytic degradation of organic contaminants by $TiO_2$. The molecular structures of the organic contaminants are represented by molecular images, which are subsequently encoded by feeding into a special convolutional neural network (CNN), EfficientNet, to extract the critical structural features. The extracted features in addition to five other experimental variables were input to a neural network that was subsequently trained to predict the photodegradation reaction rates of the organic contaminants by $TiO_2$. The results show that this machine-learning (ML) model attains a higher accuracy to predict the photocatalytic degradation rate of organic contaminants than a previously developed machine-learning model that used molecular fingerprint encoding. In addition, the most relevant regions in the molecular image affecting the photocatalytic rates can be extracted with gradient-weighted class activation mapping (Grad-CAM). This interpretable machine-learning model, leveraging the graphic interpretability of CNN model, allows us to highlight regions of the molecular structure serving as the active sites of water contaminants during the photocatalytic degradation process. This provides an important piece of information to understand the influence of molecular structures on the photocatalytic degradation process.

**Keywords:** interpretable machine learning; convolutional neural network (CNN); molecular image; photocatalytic degradation rate constant; photocatalysis

## 1. Introduction

Recently, the utilization of machine learning has been expanded to environmental engineering, such as health monitoring, wastewater forecast, environmental catalysis, etc. [1–6]. Machine-learning models have become an inexpensive but effective tool for the investigation and prediction of features of different environmental chemicals [7–10]. Particularly in the catalysis field, to accelerate new catalyst-discovery practice, data-driven machine-learning models have been used to predict catalyst properties, e.g., band gaps, and performance, e.g., adsorption energy, water-splitting efficiency, and water-contaminant degradation efficiency [11–15].

However, the "black box" character in machine learning makes the models indescribable and intractable on the relationship between the input and output. A machine is given data and an algorithm to learn, but the learning process of detecting data patterns is usually too complicated for any human to fully understand [16]. This potential uncertainty leads to raised distrust of model deployment, although some trained models have been reported with accuracy as high as 99 percent [17]. Previously, machine-learning models were usually used in low-stakes situations, such as website search and digital marketing, where explanations are not necessarily required and individual decisions do not significantly affect daily lives [18–20]. With the rapid development of artificial intelligence technologies, researchers and scientists have explored ways to implement machine learning on high-stakes decisions such as medical decisions, law, and education policy, which will deeply affect people's everyday lives [21–24]. Computer scientists have sent a warning alert on the widespread use of black-box predictive models to assist high-stakes decision making [25]. These uninterpretable models could give misleading guidance to the practice and cause more problems to society [25]. Therefore, awareness of the bias and fairness of the black-box models has been raised and the interpretability, in addition to accuracy, of the predictive machine-learning models is highly demanded [26,27].

Deep-learning models are nonlinear systems, which are more challenging to interpret than other linear models [28]. In this work, an interpretable deep neural-network model was developed that can visualize the active sites of organic contaminants during their photocatalytic degradation processes in addition to merely predicting degradation-rate constants. Here, Degussa $TiO_2$ was used as model photocatalyst for all the photocatalytic degradation processes. The photocatalytic degradation rate constants of a variety of water contaminants with presence of Degussa $TiO_2$ in aqueous phase were collected from published research articles under different experimental conditions. Molecular images were used to translate organic contaminants into machine-readable language. Combined with other experimental variables as model inputs, e.g., ultraviolet light intensity, initial contaminant concentration, initial pH of the solution, photocatalyst dosage, and experimental temperature, a deep neural network was trained and the photocatalytic degradation-rate constants were predicted. Furthermore, the most relevant regions that generate the rate constants' predictions were extracted from the molecular images through gradient-weighted class activation mapping (Grad-CAM). The highlighted regions are interpreted as the active sites of water contaminants during the photocatalytic degradation process. The predicted active sites are compared with experiments, which confirms the interpretability of the novel neural-network model. This interpretable model demonstrates the reliability of the deep neural-network models for prediction of the photocatalytic performance and to somehow open up the black box of adaptive decision making in environmental catalysis assisted by machine learning.

## 2. Results

### 2.1. Model Performance and Comparison

The model was trained and tested using a 5-fold cross-validation method. Cross-validation has advantages of full use of data and better estimation of model quality when the dataset is small. The data are firstly randomly split into five equal-sized subgroups. Four subgroups are selected for training data and the remaining one was for testing data. This process repeats five times to assure that every subgroup has been tested. The model quality is evaluated by averaging the five modeling results of each training–testing combination. The coefficient of determination ($R^2$), root-mean-square error (RMSE), and mean absolute error (MAE) are used to assess the model performance. Three models are trained and compared: CNN model trained with augmented dataset (CNN_Aug), CNN model trained with original dataset (CNN_Ori), and ANN model trained using molecule fingerprints (ANN) published in ref [15]. The scatter plots of the predicted vs. experimental photocatalytic degradation-rate constants $-\log(k)$ are shown in Figure 1. The $R^2$, MAE,

and RMSE of each testing subgroup and their averaged scores using the cross-validation method are listed in Table 1.

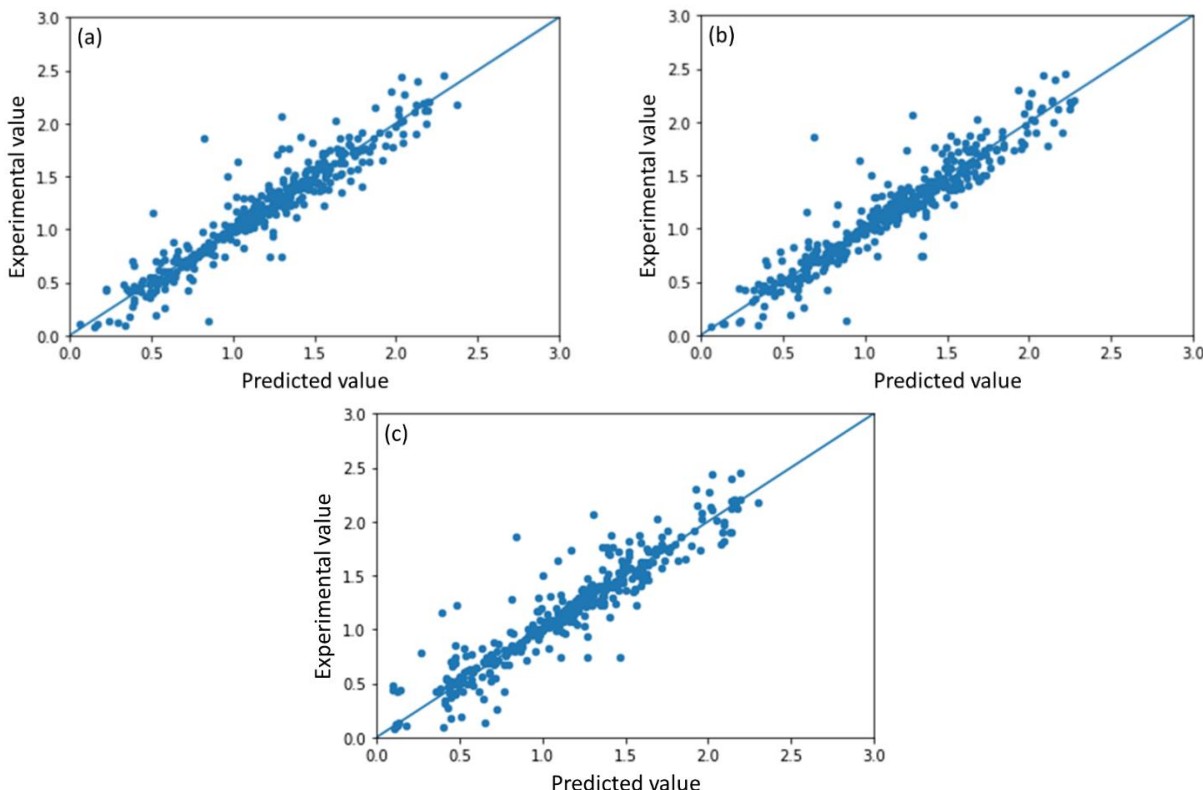

**Figure 1.** The scatter plot of the predicted vs. experimental photocatalytic degradation-rate constants $-\log(k)$ using (**a**) CNN_Aug model, (**b**) CNN_Ori model, and (**c**) ANN model.

**Table 1.** The $R^2$, MAE and RMSE of each cross-validated subgroup and their averaged scores.

| Model | Metric | Subgroup 1 | Subgroup 2 | Subgroup 3 | Subgroup 4 | Subgroup 5 | Average |
|---|---|---|---|---|---|---|---|
| | $R^2$ | 0.909 | 0.925 | 0.842 | 0.923 | 0.884 | 0.897 |
| CNN_Aug | MAE | 0.102 | 0.088 | 0.116 | 0.085 | 0.102 | 0.099 |
| | RMSE | 0.149 | 0.141 | 0.197 | 0.125 | 0.161 | 0.156 |
| | $R^2$ | 0.893 | 0.934 | 0.812 | 0.915 | 0.886 | 0.889 |
| CNN_Ori | MAE | 0.113 | 0.080 | 0.131 | 0.093 | 0.107 | 0.105 |
| | RMSE | 0.161 | 0.132 | 0.214 | 0.131 | 0.160 | 0.163 |
| | $R^2$ | 0.882 | 0.902 | 0.799 | 0.919 | 0.868 | 0.873 |
| ANN | MAE | 0.110 | 0.101 | 0.134 | 0.084 | 0.111 | 0.108 |
| | RMSE | 0.170 | 0.162 | 0.222 | 0.128 | 0.172 | 0.173 |

The CNN_Aug model has better prediction performance than the other two deep-learning models. The $R^2$ of the CNN_Aug model (0.897) is higher than that of the CNN_Ori model (0.889) and ANN model (0.873), showing better fitting accuracy. The MAE and RMSE of the CNN_Aug model (0.099 and 0.156) are lower than those of the CNN_Ori model (0.105 and 0.163) and ANN model (0.108 and 0.173), showing smaller prediction error. Both CNN models have improved performance than the ANN model, which implies that the CNN model can extract more useful features from molecular images than molecule fingerprints, resulting in smaller prediction error. Meanwhile, the CNN_Aug model has slightly better performance than the CNN_Ori model, which demonstrates that the augmented dataset

can help the model to better recognize different contaminants and extract more important features. The segmented dataset improves the model accuracy and generalizability.

To investigate how different structures of contaminants affect the prediction accuracy by the CNN_Aug model, 78 distinct contaminants were classified into six subsets according to their functional groups, i.e., amine, amide, carboxylic acid, ether, halogen, and phenyl. The prediction results of each subset are shown in Figure 2. Table 2 summarizes the $R^2$, MAE, and RMSE of each subset. It is observed that the CNN_Aug model achieved reasonable accuracy in the prediction of different groups of contaminants, among which carboxylic acid, halogen, and ether groups have better accuracy with $R^2$ around 0.9. Compared with our previous paper using the ANN model, all of the functional groups have improved accuracy. This also verifies the advantage of using the contaminant image as input and confirms the reliability of the CNN_Aug model.

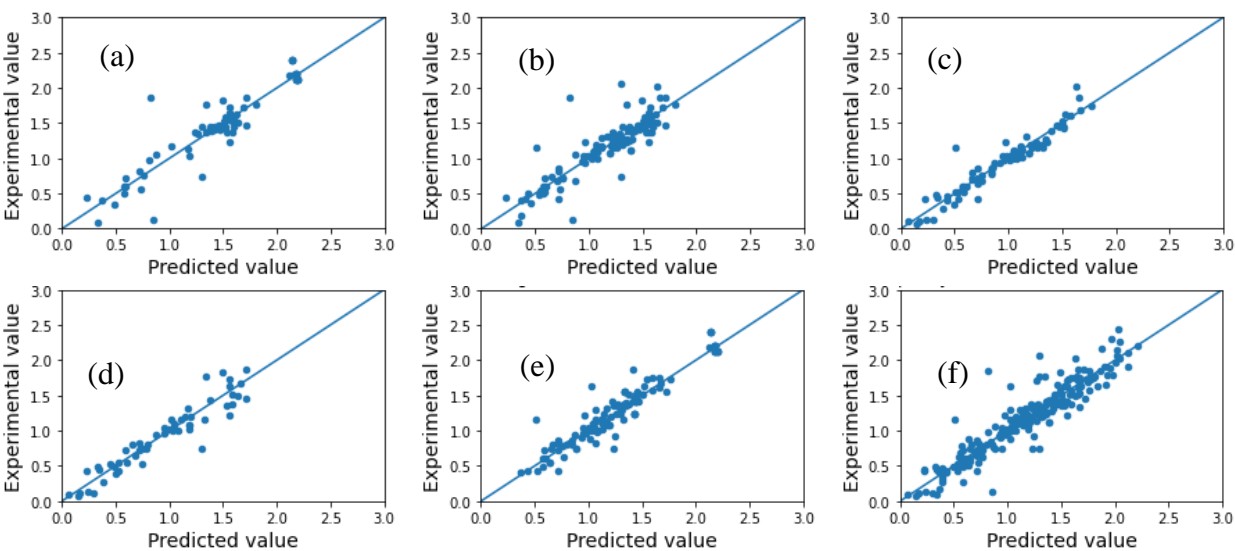

**Figure 2.** The scatter plots of the predicted vs. experimental photocatalytic degradation-rate constants $-\log(k)$ with different functional groups. (**a**) Amide, (**b**) amine, (**c**) carboxylic acid, (**d**) ether, (**e**) halogen, and (**f**) phenyl.

**Table 2.** The $R^2$, MAE and RMSE of different functional groups.

| Metric | Amide | Amine | Carboxylic Acid | Ether | Halogen | Phenyl |
|---|---|---|---|---|---|---|
| $R^2$ | 0.868 | 0.756 | 0.917 | 0.891 | 0.905 | 0.869 |
| MAE | 0.127 | 0.128 | 0.079 | 0.114 | 0.101 | 0.118 |
| RMSE | 0.210 | 0.209 | 0.124 | 0.156 | 0.151 | 0.184 |

### 2.2. Feature Importance

The effects of input variables on the degradation-rate constants were analyzed to evaluate the impact of the photocatalytic-reaction parameters by using the SHAP method [29]. It assigns each feature an importance value, i.e., SHAP value, for a particular prediction indicating how much a model prediction relies on each feature; in other words, how much each feature contributes to the prediction. Moreover, it is capable of identifying the impact of each feature on overall model prediction, i.e., the positive and negative relationships of the features with the degradation-rate constants. This assists the understanding of the impacts of input variables on the predicted outcome. In order to compare the importance of parameters fairly, we set the same initial weight for each parameter. Although the contaminant occupies 1280 nodes, it is treated as a single parameter. To avoid weakening the importance of other experimental parameters, each of the initial weight of 1280 nodes of

the contaminant type was divided by 1280. Figure 3 shows the SHAP value densities of the five experimental variables. The water-contaminant type is excluded because it is categorically dated and it is difficult to define any positive or negative impact on degradation-rate constants.

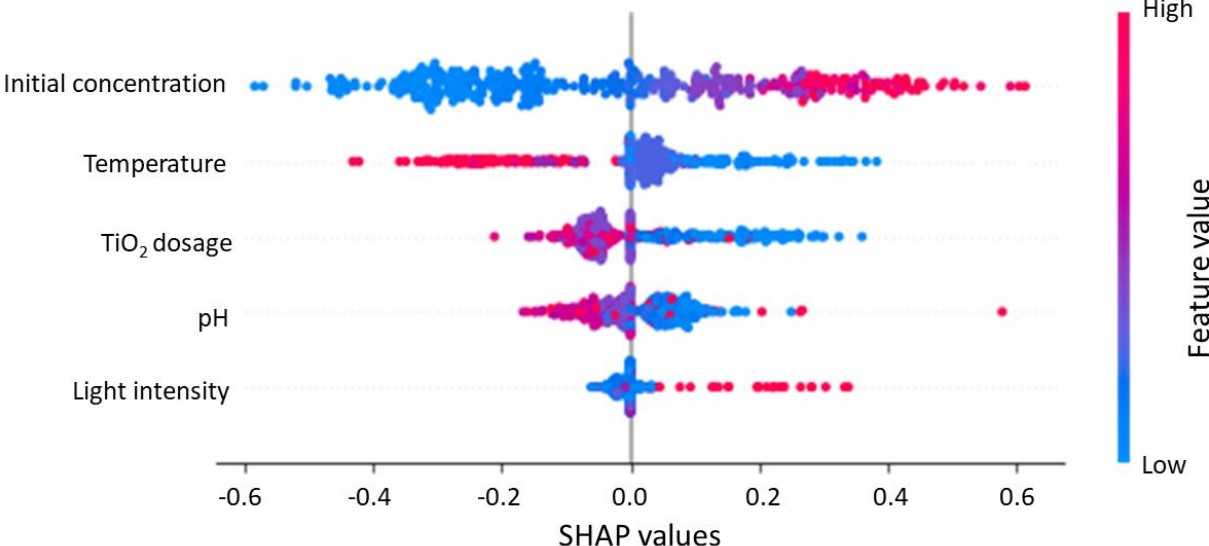

**Figure 3.** Density scatter plot of SHAP values of the input variables for the prediction of degradation-rate constants.

In the row of each input, the points indicate the corresponding SHAP values with the magnitude measured by horizontal ordinate and the color represents the input value of that individual (the red code means a high value and the blue code means a low value). Meanwhile, the points on the left side of the midline indicate the negative effect on the predicted rate constants $-\log(k)$, while the points on the right side indicate the positive effect. The farther away from the midline, the greater the absolute impact on the predictions. For example, the lower values of initial concentration (blue points) are mainly distributed on the left side while the higher values (red points) are mainly distributed on the right side, indicating that higher initial concentration tends to increase model output $-\log(k)$. Since the output $-\log(k)$ has a negative sign, theoretically it means a higher initial concentration leads to lower the rate constant k, which is consistent with the experimental observations [30–32]. Similarly, temperature has a negative effect on model output $-\log(k)$, which indicates that higher temperature tends to increase the rate constant k. Higher $TiO_2$ dosage tends to increase the rate constant k, since most red dots are on the left and blue dots are on the right, but it is also observed that some red dots are scattered on the right side. In experiment, it is found that the rate constant is directly proportional to the catalyst amount when the catalyst amount is relatively small. After catalyst loading reaches a limit, the rate constant slightly decreases [33]. This explanation matches the model prediction and the red dots observed on the right side could be the $TiO_2$ dosage beyond the loading limit. As for pH, our model prediction shows that pH has a dominantly positive impact on the increase in the rate constant despite some individual dots performing oppositely, because most of the red dots are found on the left and most of the blue dots are on the right while some red dots are observed on the right and some blue dots are also found on the left. In reality, it is not necessarily true that higher pH will always increase the rate constant, because pH variations can positively or negatively affects the adsorption of the organic molecules on the catalyst surface due to different molecular structures [33]. Our model indicates that higher pH, an alkaline environment, will increase the rate constant for the majority of the contaminants analyzed, and for the other contaminants, an acidic environment contributes to a faster degradation reaction. As for light effect, the model shows that higher light intensity tends to decrease the rate constant, which is not consistent

with the experiment. This might be due to the relatively small light effect and the errors from the light-intensity measurement.

The distance between SHAP point and midline indicates the feature importance. By averaging all of the SHAP values of each feature, we can obtain the overall feature importance of each input variables, as shown in Figure 4. The importance factor of water contaminant is obtained by adding the SHAP values of contaminant feature vector (vector of 1280 generated by EfficientNet). Figure 4 also compares the importance factors of the CNN_Aug model in this work and the previous ANN model [15]. The feature importances of the two models share the same orders: contaminant type > initial concentration > temperature > TiO$_2$ dosage > pH > light intensity. The corresponding importance factors are also in the same order of magnitude except for the contaminant type, which is reduced from 1.024 to 0.291 by using the CNN_Aug model. This could be due to the molecular expression of the contaminants for the machine to process. In the ANN model, molecular fingerprints with a binary vector of 512 were used to represent the molecule while the CNN_Aug model transforms the molecular image into 200 × 200 pixels, which is in total 40,000 data points. In addition, among those 40,000 data points, many of them are blank information. Therefore, the CNN_Aug model has a denser molecular expression with more blanks. This weakens the importance of contaminant type and improves the influence of other reaction conditions on the degradation-rate constant.

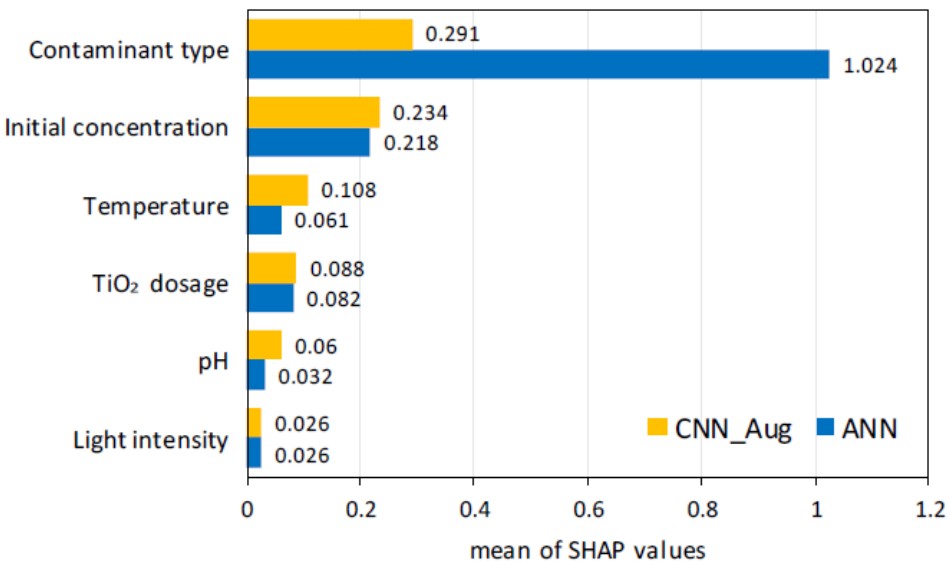

**Figure 4.** The bar plot of the importance factors of each input variable.

### 2.3. Interpretability of Active Site Prediction

A good deep-learning model should not only achieve a high accuracy, but also have a reasonable interpretability. To better interpret the prediction, we extracted the useful features from molecular images and rebuilt the heat maps for visual explanation of photocatalytic degradation process. Gradient-weighted class activation mapping (Grad-CAM) is used to extract the most relevant regions in the molecular image that make the prediction [34]. It is a class-discriminative localization technique that generates visual explanations for any CNN-based network without requiring architectural changes or retraining. The important regions of the image which correspond to any decision of interest are visualized in high-resolution detail, making the CNN model more transparent and explainable. The process of Grad-CAM is shown in Figure 5.

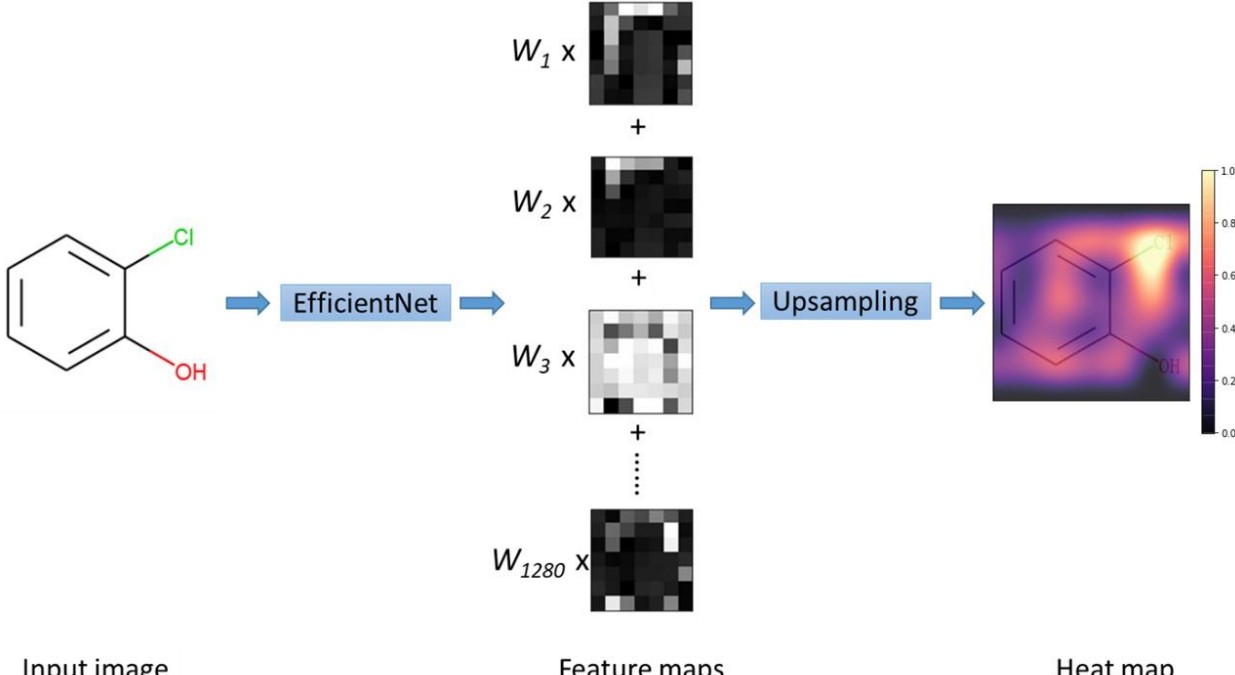

**Figure 5.** Process of Grad-CAM.

Given a molecular image as input, we first forward-propagated the image through the convolution layers of the EfficientNet model to extract the output of the last convolutional layer, i.e., feature maps. Feature maps contain the richest information of an image. Then, we aggregated all feature maps according to corresponding weight (W). The weight of the feature map can be calculated with the weights of each pixel. Finally, we upsampled the aggregated map to the same size as the input image to obtain the heatmap, which represents the important region that the model extracts. The heat map in Figure 5 shows the highlighted area of 2-chlorophenol and "−Cl" are recognized. In theory, "−Cl" is the active site involved in the photocatalytic degradation process of 2-chlorophenol. Some examples of heat maps of other water contaminants are shown in Figure 6. We also searched for the active sites from the photocatalytic degradation pathways in experiments. The predicted active sites from the heat maps and experiments were compared to investigate the capability of the heat maps to capture the active sites involved in the photocatalytic degradation process. In Figure 6, the experimental part gives the possible molecule transformation in the first step of the photocatalytic degradation process [35–40]. The functional groups are substituted or decomposed after being attacked by active oxygen species (●OH, ●$O^{2-}$, and $H_2O_2$) and these functional groups are referred as active sites in the heat maps. The findings by the heat maps are reasonable and consistent with the degradation pathways from experiment. These observations verify that the model can properly identify and extract the important features of molecular images contributing to the prediction.

However, some irrational heat maps were observed, and some examples are provided in Figure 7. Essentially, three types of irrational heat maps were found. Figure 7a shows that the highlighted regions of n-propanol are all located in the blank area. Figure 7b is the heat map of acetone that highlights the majority of the image. Figure 7c highlights all the functional groups of benzylparaben. It is believed that the model uncertainty and errors come from these irrational heat maps. The unreasonable mapping could be due to two reasons. First, the molecular images are decomposed to 200 × 200 pixels, which also include the blank areas. Pixels that contain the chemical information and the blank are both processed by the model. Theoretically, the blank in molecular image does not have a meaning in the photocatalytic reactions; however, they are given a definition and used as training data when transforming the image information by the model. This makes

the model confused about the blank and the real chemical information, so some of the predictions give irrational mapping. Second, all of the molecules were transformed to the same image size in despite of their different molecular sizes. Therefore, the molecular images exhibit different sizes of bonding widths, or chemical fonts. For example, as shown in Figure 6a,e, 2-cholorophenol and dichlorvos both have halogen group "−Cl". Given the differences in molecular size, the same "−Cl" group appears slightly different in molecular images. This could increase the difficulty for the model to identify the same functional groups, and raise the uncertainty and inaccuracy to the model.

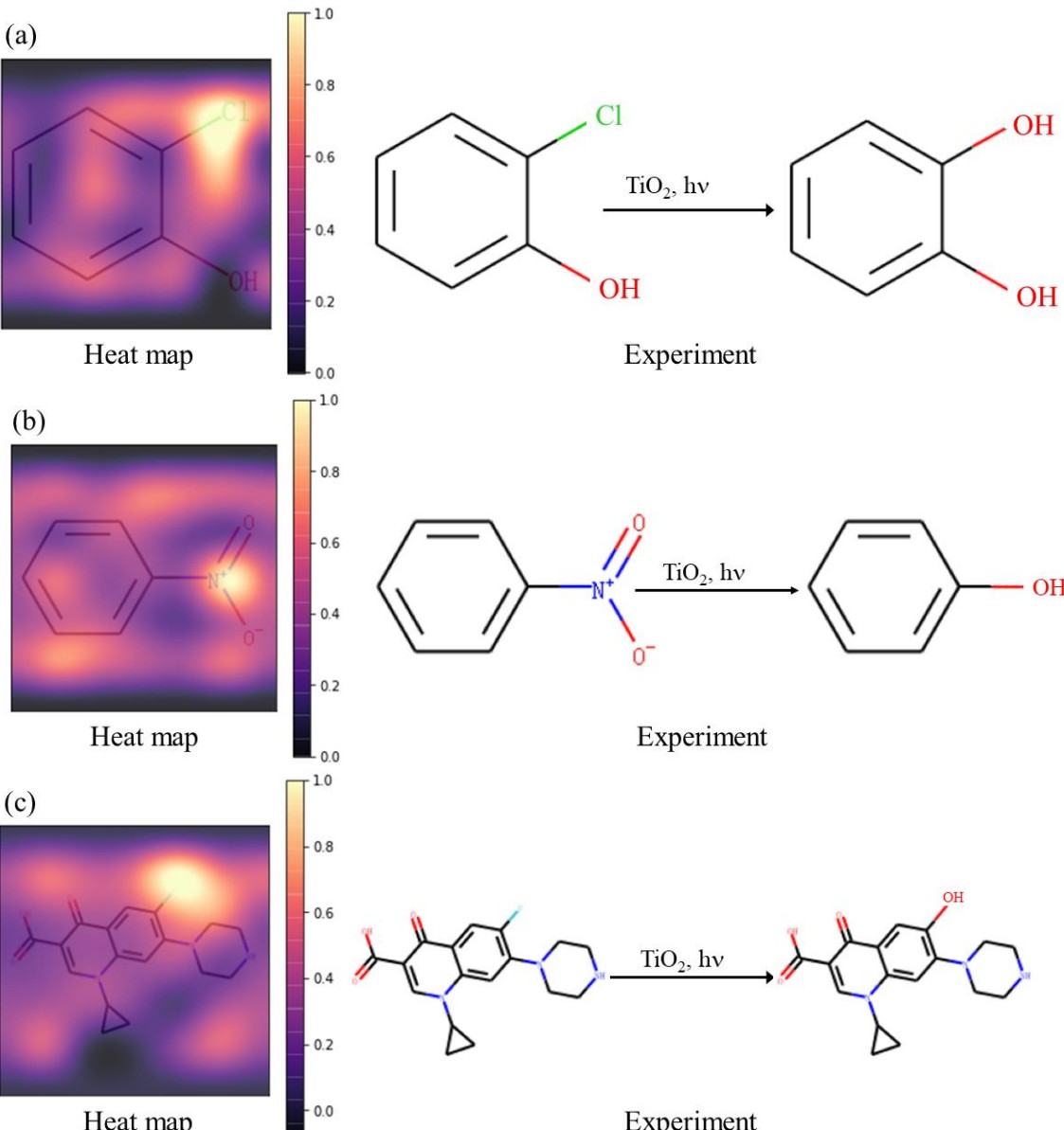

**Figure 6.** *Cont*.

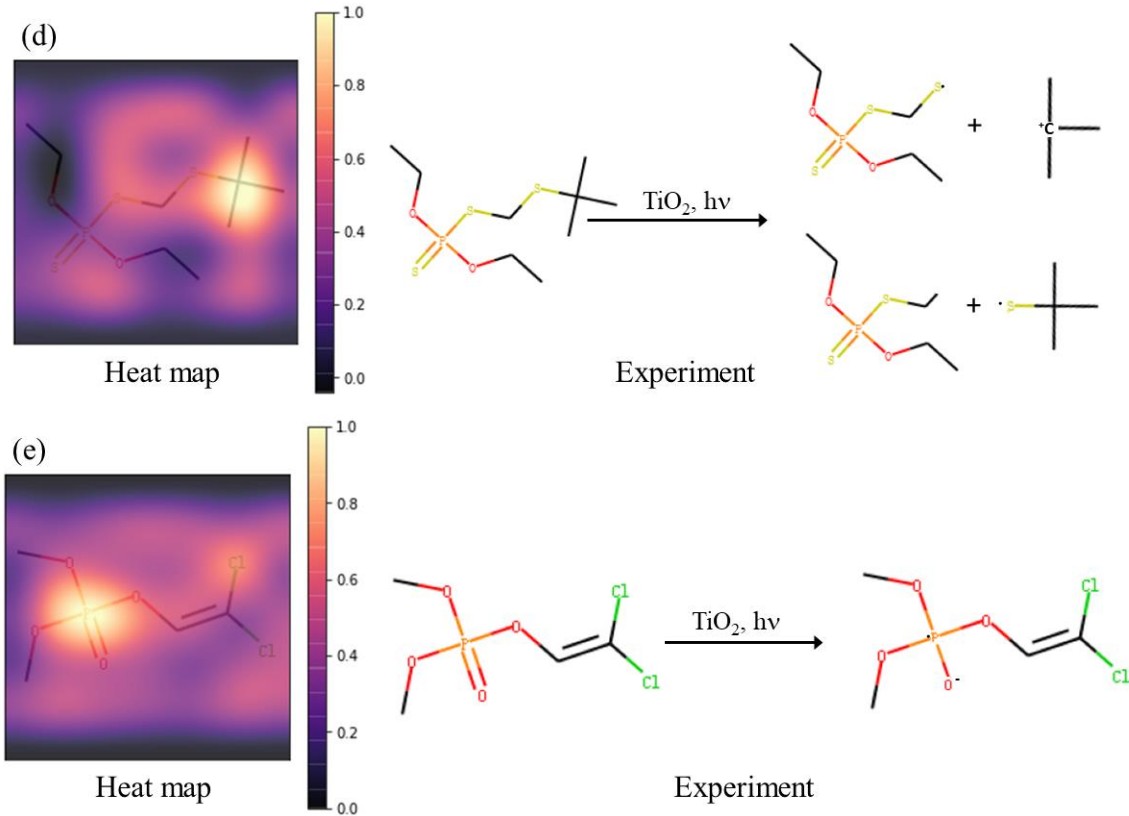

**Figure 6.** Predicted heatmaps and comparison of active sites with experiments. (**a**) 2-Cholorophenol, (**b**) nitrobenzene, (**c**) ciprofloxacin, (**d**) terbufos, and (**e**) dichlorvos.

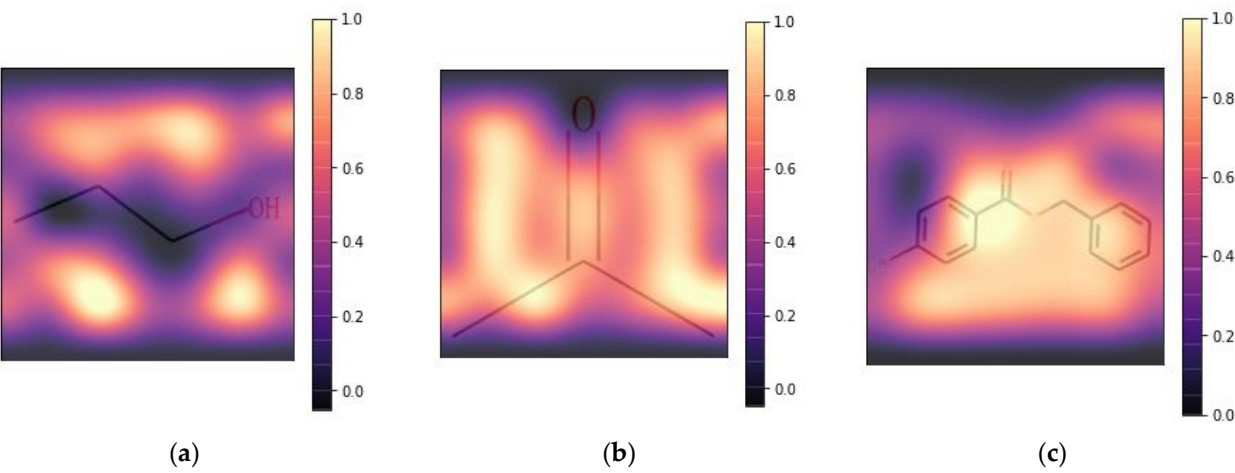

**Figure 7.** Examples of heat maps with bad prediction. (**a**) n-Propanol, (**b**) acetone, and (**c**) benzylparaben.

Approximately 30% of the heat maps are irrational maps that little chemical information can be extracted from the highlighted regions. It is also observed that chemicals with larger molecular weight have a higher percentage of irrational maps. This might be due to the fact that each feature of chemicals with larger molecular weight occupies less pixels in the molecular images. Therefore, the importance of each feature is depressed, which makes the model difficult to predict based on different chemical features.

## 3. Methods

### 3.1. Datasets

A dataset of 446 data points of photocatalytic experiments with $TiO_2$ for training and testing models was collected from the published research articles. The details of the dataset are described in our previous published paper [15]. The input variables are organic water-contaminant type, initial concentration of water contaminant (mg $L^{-1}$), initial pH of the solution, $TiO_2$ dosage (g $L^{-1}$), ultraviolet light intensity (mW $cm^{-2}$) and experimental temperature (°C). The predicted output is the photocatalytic degradation-rate constant of the water contaminant ($min^{-1}$). Degussa P25 $TiO_2$ nanopowder is used as photocatalyst from all experiments.

The water contaminants are nonnumerical variables that are required to be converted to a computer-readable language. In this work, the input of the water-contaminant type is represented as a molecular image. RDKit, an open-source cheminformatics for machine learning, was used to encode chemical formulas into molecular images [41]. The conversion process involves two steps: the chemicals are firstly converted to SMILES (simplified molecular-input line-entry system) strings, and then further transformed to molecular images. SMILES strings are line notations for organic molecules that describe their atomic information and structures. Polymorphs can be differentiated by SMILES strings, which place the branch from a chain directly after the atom to which it is connected. For example, n-propanol and isopropanol have the same chemical formula, $C_3H_8O$. Their SMILES strings are CCCO and CC(O)C, respectively, which leads to distinct molecular images. After conversion, the size of the molecular image is set as 200 by 200 pixels. Figure 8 is an example of a water contaminant (benzoic acid) converting to the molecular image.

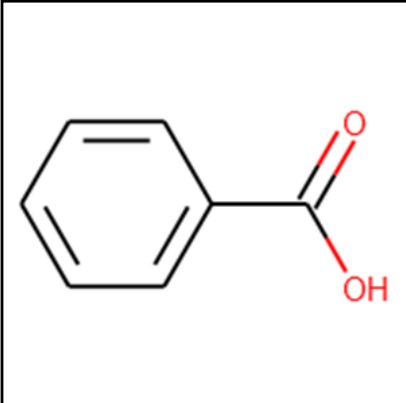

**Figure 8.** The molecular image of benzoic acid.

### 3.2. Data Augmentation

The drawback of encoding chemical formula into molecular image is the lack of accuracy in recognizing the same contaminants with different orientations. Specifically, the image looks different but represents the same original contaminant if it is rotated or flipped. Therefore, the model may fail to recognize the same contaminant with a rotated or flipped image if the images with the original orientation alone are trained. For this purpose, data augmentation was applied to improve the image identification. Through rotation and flip operation, eight images for each contaminant can be generated. Figure 9 shows the image-augmentation operations for benzoic acid. As a result, the size of training dataset is increased by eight times.

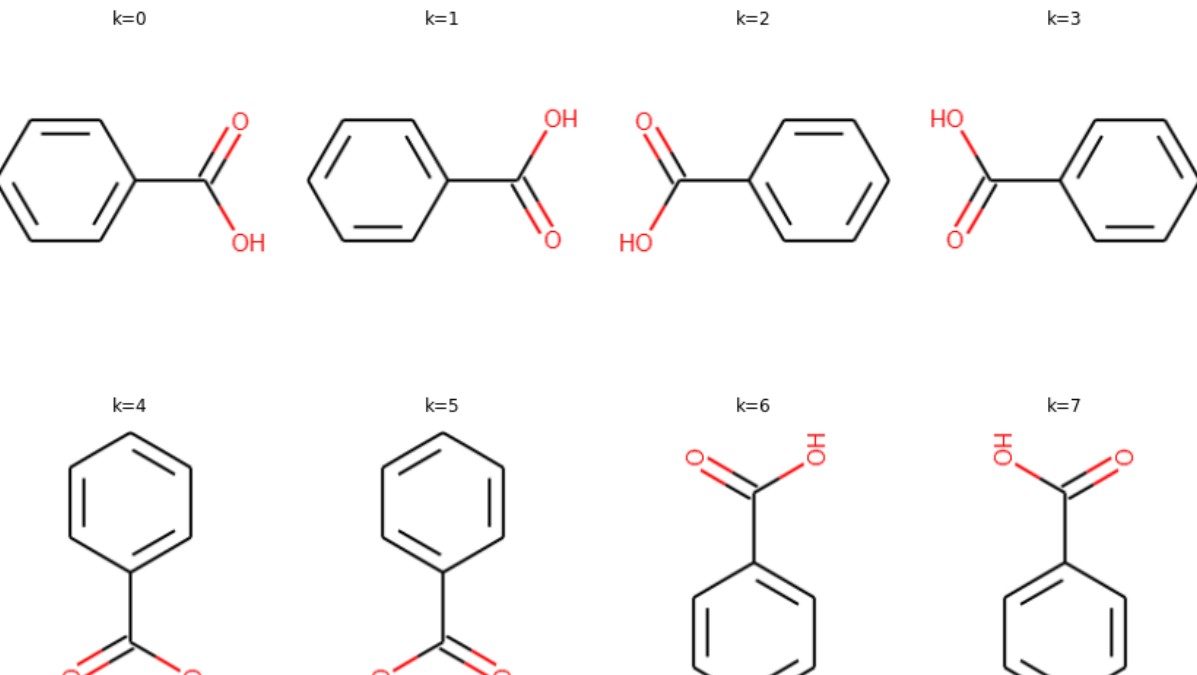

**Figure 9.** Rotation and flip image of benzoic acid.

### 3.3. Model Structure

A convolutional-neural-network (CNN) model was developed to predict the first-order photocatalytic degradation-rate constant k. The model structure is shown in Figure 10. The CNN model is capable of extracting useful features from molecular images. The extracted features in addition to other five input variables were processed by the neural network to predict the rate constants. The convolutional layers for image feature extraction are referred to EfficientNet, which was created by Google in 2019 [42]. EfficientNet reduces parameter size and computation by an order of magnitude, which demonstrates higher accuracy and better efficiency over existing CNNs. The output of EfficientNet is a one-dimensional vector of 1280 neurons. 1280 is the optimal number of neurons, which was searched as a hyperparameter by Bayesian optimization algorithm. In addition to other five experimental variables, the feature layer has 1285 neurons. The fully connected layer has 256 neurons and the output layer has one neuron representing the predicted rate constant.

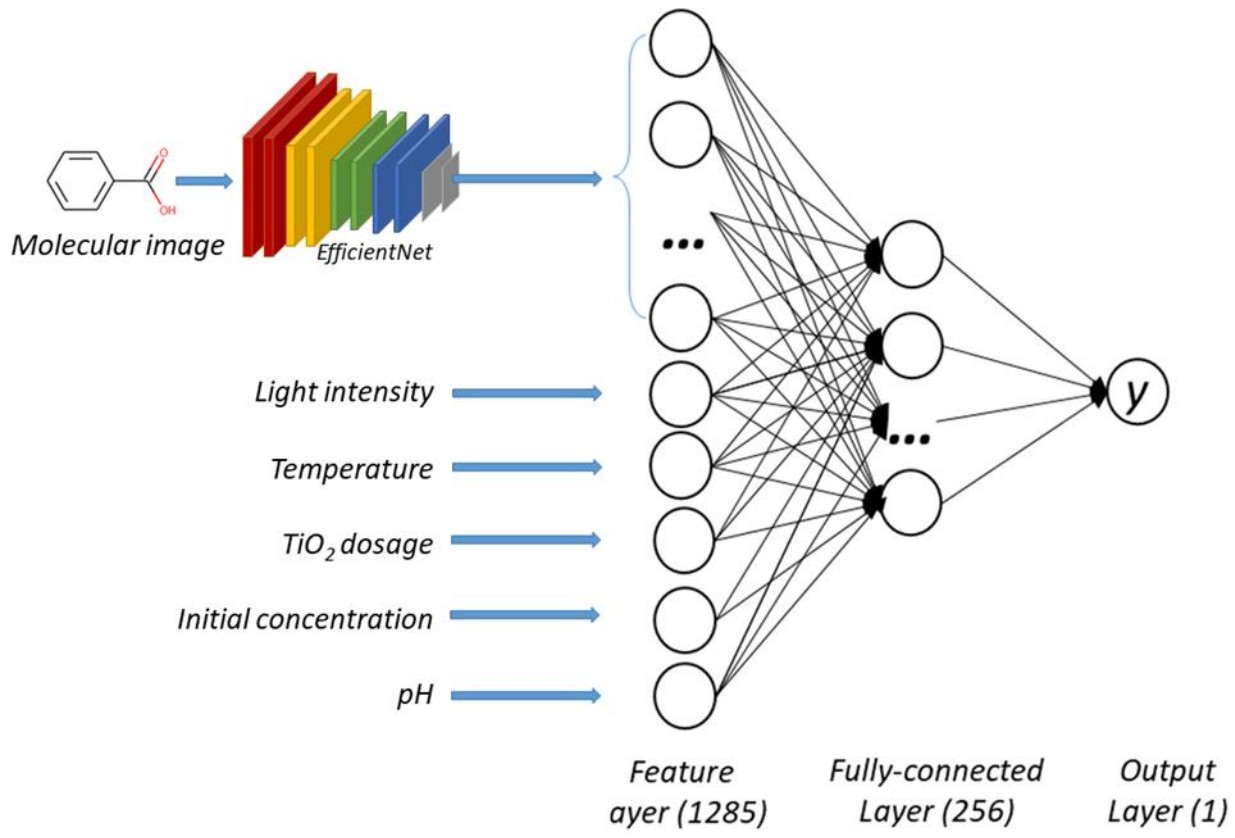

**Figure 10.** The CNN model structure for the photocatalytic degradation-rate constant prediction.

## 4. Conclusions

An interpretable convolutional neural-network model, MI-CNN, was developed by use of molecular images to encode water contaminants to predict the photocatalytic degradation-rate constants of the contaminants with the presence of Degussa $TiO_2$. The model achieved improved performance over molecular fingerprint encoding in terms of prediction accuracy. Furthermore, using gradient-weighted class activation mapping, this model is able to create heat maps that capture the most relevant regions in the molecular images that generate the prediction. These highlighted regions can be interpreted as the active sites in the organic molecules that are involved in the photocatalytic degradation process. After carefully comparing the predicted active sites with the degradation pathways from experiments, it confirms that some functional groups attacked by active oxygen species ($\bullet OH$, $\bullet O^{2-}$, and $H_2O_2$) are correctly captured in heat maps. These visual interpretations of heat maps add transparency and confidence to the model prediction, which somehow opens up the black box of predictive deep-learning models in the application of environmental photocatalysis.

## 5. Discussions

The major contributions of this paper are in the development of an interpretable machine-learning (ML) model to predict the performance of photocatalytic degradation of organic contaminants by $TiO_2$. This ML model integrates the convolutional neural network (CNN) with the artificial neural network (ANN). It utilizes molecular images to encode the molecular structures of the organic contaminants, whose features are extracted by EfficientNet, a special type of CNN model. The extracted features together with other experimental variables affecting the photocatalytic reaction rate were input to an artificial neural network. The cascaded CNN and ANN model is subsequently trained to predict the photodegradation reaction rates of the organic contaminants by $TiO_2$.

The results show that this machine-learning (ML) model achieved a high accuracy to predict the photocatalytic degradation rate of a wide range of organic contaminants. Heat maps can be generated to identify the most relevant regions in the molecular image affecting the photocatalytic rates by use of gradient-weighted class activation mapping (Grad-CAM). This provides an important piece of information to understand the influence of molecular structures on the photocatalytic degradation process.

While the paper used $TiO_2$ to demonstrate the general procedures, the methodology can be extended to other types of photocatalysts.

**Author Contributions:** Conceptualization, X.Y.; methodology, X.Y. and J.H.; software, J.H.; validation, Z.J.; formal analysis, Z.J. and J.H.; investigation, J.H.; Z.J.; resources, X.Y.; data curation, Z.J.; writing—original draft preparation, Z.J. and J.H.; writing—review and editing, X.Y.; visualization, J.H.; supervision, X.Y. and A.S.; project administration, X.Y.; funding acquisition, X.Y. and A.S. All authors have read and agreed to the published version of the manuscript.

**Funding:** This work is supported by US National Science Foundation with an award number: 1563238.

**Data Availability Statement:** Data are available by request from the correspondence author.

**Acknowledgments:** The authors acknowledge the support of US National Science Foundation for supporting this research.

**Conflicts of Interest:** The authors declare no conflict of interest.

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
