# Peer review of "Predicting Active Sites in Photocatalytic Degradation Process Using an Interpretable Molecular-Image Combined Convolutional Neural Network"

_catalysts, doi:10.3390/catal12070746_

Round 1

Reviewer 1 Report

Please find the attach file

Author Response

Response to reviewer #1:

  1. The abstract must be rewritten. The abstract should be state briefly the purpose of the research, the principal results and major conclusions.

 Response:  The abstract has been rewritten according to the comment.

  1. The scientific contribution of this work is not clear, it is strongly recommended to present a table that summarizes the bibliographic review and highlights the real contribution of this research compared to other articles.

 Response:  The major contributions of this paper is in the development of an interpretable machine learning (ML) model to predict the performance of photocatalytic degradation of organic contaminants by TiO2.   This ML  model integrate the convolutional neural network (CNN)  with artificial neural network (ANN).  The molecular structures of the organic contaminants are represented by the molecular images, which is subsequently encoded by feeding the images into EfficientNet, a special type of CNN model, to extract the critical structural features.  The extracted features in addition to other experimental variables affecting photocatalytic reaction rate were input to a neural network that is subsequently trained to predict the photodegradation re-action rates of the organic contaminants by TiO2

The results show that this machine learning (ML) model attains a higher accuracy to predict the photocatalytic degradation rate of organic contaminants than a previously developed machine learning model that used molecular fingerprints encoding.   Heat map can be generated to identify the most relevant regions in the molecular image affecting the photocatalytic rates by use of Gradient-weighted Class Activation Mapping (Grad-CAM).  This provides an important piece of information to understand the influence of molecular structures on the photocatalytic degradation process.

While the paper used TiO2 to demonstrate the general procedures, the methodology can be extended to other types of photocatalysts.

  1. In section model structure, the authors mention that “CNN model was developed to predict the first order photocatalytic degradation rate constant k.” The authors should better specify the type of CNN methods and limits they have applied to be clarify for the readers. The type of ANNs explored in this works is somewhat limited. Other types of ANN might have better performance in this particular task.

 Response:   A unique ML model structure is developed in this work.   The molecular information of organic contaminants is represented by its molecular image.  A special type of CNN model, EfficientNet, takes the molecular images as inputs and extract the critical features of the molecular images, which is represented as a 1024 vector (after hyperparameter optimization).   This information is then fed as inputs (together with other experimental variables that affecting the photocatalytic reaction rate) into the ANN model structure. Details are introduced in the model structure session, which also include relevant reference to EfficientNet etc.   

The unique advantage of this ML model is that heat map can be generated by use of Gradient-weighted Class Activation Mapping (Grad-CAM) to identify the most relevant regions in the molecular image affecting the photocatalytic rates.   This significantly enhance the interpretability of photocatalytic reaction mechanism during the photocatalytic degradation of organic contaminants. 

  1. The authors should provide a justification for the low value of the coefficient of determination (R2) and high values of root mean square error (RMSE) and mean absolute error (MAE) for the rate constants –log(k)

 Response: The R2 is about 0.9, which is not very low for a predictive machine learning model. However, some other published predictive models for prediction of reaction rate, including Generic programming (GP), Response surface methodology (RSM), and deep neural networks, have a higher R2, usually between 0.97 and 0.99. And our model has a high RMSE and MAE, which leads to about 10 to 20% error for rate constant k. The lower R2 and higher RMSE of our model might be due to a wider range of chemicals incorporated in our model. Totally 76 organic contaminants with very different structures and features were trained and tested. Majority of the previous models only considered a single type of contaminant, and a few models included several types of contaminants with a similar chemical structure. Therefore, our model is more generalized but the accuracy is somehow sacrificed.

  1. This paper describes the results but little discussion is included, therefore the authors should improve notably the discussion of their results to show the scientific contribution.

 Response:  Besides to further refine the discussion the implications of the ML model prediction. A section titled Discussions has been added at the end of the paper to highlight the major contributions of this work, as well as the generality.

The major contributions of this paper is in the development of an interpretable machine learning (ML) model to predict the performance of photocatalytic degradation of organic contaminants by TiO2.   This ML model integrate the convolutional neural network (CNN)  with artificial neural network (ANN).  It utilizes the molecular images to encode the molecular structures of the organic contaminants, whose features are extracted by EfficientNet, a special type of CNN model.   The extracted features together with other experimental variables affecting the photocatalytic reaction rate were input to an artificial neural network.   The cascaded CNN and ANN model is subsequently trained to predict the photodegradation re-action rates of the organic contaminants by TiO2

The results show that this machine learning (ML) model achieved a high accuracy to predict the photocatalytic degradation rate of a wide range of organic contaminants.   Heat map can be generated to identify the most relevant regions in the molecular image affecting the photocatalytic rates by use of Gradient-weighted Class Activation Mapping (Grad-CAM). This provides an important piece of information to understand the influence of molecular structures on the photocatalytic degradation process.

While the paper used TiO2 to demonstrate the general procedures, the methodology can be extended to other types of photocatalysts.

  1. Future scope of work and the scientific reason behind these results should be added in the conclusion section.

Response:   The conclusion section has been further refined.  Besides, a new section titled Discussion has been added to the end of the revised paper.  It provides a summary of this work and also stated the implications for developing ML based photocatalytic performance prediction for other types of photocatalysts.

Round 2

Reviewer 1 Report

After reviewing the correction made by the authors in the original version  the paper. I recommend to be accepted this manuscript.